# The Relationship between Green Organization Identity and Corporate Environmental Performance: The Mediating Role of Sustainability Exploration and Exploitation Innovation

**DOI:** 10.3390/ijerph16060921

**Published:** 2019-03-14

**Authors:** Xinpeng Xing, Jianhua Wang, Lulu Tou

**Affiliations:** 1School of Business, Jiangnan University Wuxi 214122, China; xinpeng@jiangnan.edu.cn (X.X.); 1080216411@vip.jiangnan.edu.cn (L.T.); 2Food Safety Research Base of Jiangsu Province, Jiangnan University, Wuxi 214122, China

**Keywords:** green organization identity (GOI), sustainability exploration innovation (SER), sustainability exploitation innovation (SEI), government environmental regulation (GER)

## Abstract

The link between green organizational identity (GOI) and corporate environmental performance (CEP) has been investigated, but existing studies have no consistent conclusion. A significant research gap remains regarding the mediating role of sustainability exploration innovation (SER), sustainability exploitation innovation (SEI), and the moderating role of government environmental regulation (GER). This study explored the relationship between GOI and CEP in a moderated meditation model which includes SER, SEI, and GER. Using structural equation modelling and bootstrap method based on data sets from of 380 Chinese companies, the results show that: (1) GOI promotes SER, thereby enhancing CEP; (2) GOI promotes SEI, thereby enhancing CEP; (3) GER can positively moderate the indirect effect of GOI on CEP via SER; (4) GER negatively moderate the indirect effect of GOI on CEP via SEI. These findings suggest that firms choose different innovative ways between SER and SEI to improve CEP which depends on different levels of GER in China.

## 1. Introduction

In recent years, how to simultaneously achieve high-quality economic development and eco-friendly ecological environment have received increasing amounts of attention. Specially, ecological environmental protection become the major challenges in the period of China’s economic transformation [1]. However, as the micro-subject of economic development and ecological environmental protection, the balance between enterprise development and ecological environmental protection greatly influences and determines the realization of economic development and ecological environmental protection [2]. Therefore, promoting the green development of economy and protecting the ecological environment from the micro level have become a difficult problem that the academic and practical circles need to be solved urgently.

Among various elements of corporate environmental management, green organizational identity (GOI), government environmental regulation (GER), sustainability exploration innovation (SER), and sustainability exploitation innovation (SEI) arguably become a facet where the government has always been attempting to influence firm innovation behavior to improve the effectiveness and efficiently of corporate environmental management [3,4,5]. Although a large number of studies have shown that the government needs to formulate environmental regulations to regulate the environmental management of enterprises, there is no consistent consequence how governments can formulate regulations and how enterprise manage green innovation to improve their own performance according to GER [6].

Furthermore, as one of the research topics of enterprise environmental management, green organizational identity (GOI) has been gradually valued by a large number of studies [7,8]. Currently, most existing studies focus on revealing the effect of GOI on innovation performance and green competitive advantage [9]. Some scholars who pay attention to GOI believe that the effect of GOI on organizational performance is also affected by internal factors (e.g., innovation) and external factors (e.g., environmental regulation). However, unlike the enterprises in developed countries, the focus of enterprises like emerging economies such as China is still developing rapidly, and the willingness to reduce emissions has not yet formed, and the awareness of environmental protection is slowly forming. Under such circumstances, whether the company is willing to establish a GOI, and whether GOI can effectively promote SER and SEI has not been answered. Hence, it is necessary to further explore the boundary conditions of the action path to clarify the validity of the path [10].

As two important ways of green innovation, sustainability exploration innovation (SER), and sustainability exploitation innovation (SEI) can effectively balance corporate sustainability and innovation performance [11]. Existing research showed that green innovation is one of the best ways to ease the double pressure of economic growth and environmental protection [12]. As the positive relationship between sustainability exploration innovation (SER), sustainability exploitation innovation (SEI), and green innovation examined by some previous studies [13], the existing studies have not subdivided the innovation behaviors of enterprises and explored the influence of GOI on different types of innovation behaviors, especially on SER and SEI.

In addition, GER is an external factor that can influence innovation (e.g., SER and SEI), and play an important role in environmental management. Most existing research on GER at the corporate level focuses on the “Porter hypothesis” test [6]. However, few studies have sought to explore the role of GER between GOI to corporate environmental performance (CEP). Given the proposal proposed by Li and Ramanathan [14] and Sanchez and Mckinley [15], it is necessary to reveal the mechanism of GER in corporate environmental management from different perspectives. The Chinese government attaches great importance to environmental protection, and also emphasizes accelerating economic transformation and development, but theoretical support and practical guidance are insufficient. GER is a powerful administrative means, which makes enterprises must be disciplined. In this case, it is particularly important to explore GER to regulate the environmental protection behavior of enterprises (e.g., SER, SEI, and GOI). Therefore, it is very necessary to take the Chinese enterprises as the research object and deeply explore the moderation effect of GER on the indirect effect of GOI on CEP via SER and SEI, and further test the correctness and adaptability of the existing research conclusions with a view to promoting the research of GOI, SER, and SEI in developing countries.

Hence, following the recommendation of Ramanathan et al. [4] and Maletič et al. [3], this study aims to explore the role of GER, GOI, SER, and SEI in enterprise environmental management regarding the moderation effect of GER as well as the mediation effect of SER and SEI. Therefore, this research takes Chinese enterprises as the research object, through theoretical model construction and empirical research, in order to promote and improve the research on GOI, SER, and SEI.

## 2. Literature Review and Hypotheses

### 2.1. Sustainability Exploration Innovation (SER) and Sustainability Exploitation Innovation (SEI)

Based on the exploration-exploitation paradigm and the organizational sustainability theory, Maletič et al. [11] proposed and constructed a theoretical framework for SER and SEI in 2014. In consideration of corporate sustainability, SER emphasizes new design, product, technology and knowledge to resolve the environmental cost of product lifecycle by strengthening the sustainable development-oriented learning capability, and focuses on corporate sustainability in order to realize future prosperity and build competitive advantages [12]. While SEI emphasizes the improvement of existing technology and product to effectively reduce the consumption of materials, water, and energy by improving the quality of product and service or enhancing existing functions. SEI also aims at improving corporate sustainability in the short term and existing competitive advantages [8]. Compared with traditional exploration innovation and exploitation innovation, SER and SEI aim to help enterprises to build the market, social and environmental competitive advantages, which is beneficial for the enterprises to deal with ecological, social, and economic challenges [3]. 

In the subsequent studies, Maletič et al. [12] conducted an empirical study on the effect of SER and SEI on CEP, and the results show that SER and SEI exert different impacts on CEP for enterprises in different countries. Moreover, for different companies in the same country, influences of SER and SEI on CEP are different. Subsequently, Maletič et al. [3] analyzed the influencing mechanism of SER and SEI on CEP using the data from European enterprises. It is discovered that SER and SEI also have different impacts on CEP in different scenarios. For example, when environmental competitiveness and uncertainty are at a lower level, SEI is more beneficial to CEP. On the contrary, when environmental competitiveness and uncertainty are at a relatively high level, SER proves to be beneficial for CEP. The impact of SER and SEI on CEP has not been unified. Considering the development stage and institutional problems of an emerging economy such as China, the question of whether Chinese enterprises prefer to promote CEP through SER and SEI has not been answered.

### 2.2. Green Organizational Identity (GOI)

Organizational identity can be interpreted as a set of the most primary, unique, and everlasting beliefs in an organization [7,16]. According to the organizational identity theory, the organizational identity enables members to understand the organization better, enhance their awareness of organizational goal, and help them keep pace with the organization [17]. With the enhancement in international environmental regulation and customers’ environmental awareness, organizations proactively pay attention to environmental management issues, and put enough emphasis on reducing environmental pollution by changing organizational behaviors [13]. To reconcile the conflict between organizational development and environmental protection, Chen proposed the concept of GOI which incorporated environmental protection into the framework of the organizational identity based on organizational identity [9]. The positive relationship between GOI and innovation performance has been examined by subsequent research. Chang and Chen [13] explored the relationship between GOI and green innovation performance, and argued that GOI has a direct impact on green innovation performance. Furthermore, Song and Yu [7] constructed the green innovation strategy framework featuring GOI, and discovered the mediating role of GOI between green innovation strategy and green creativity. It can be seen that the existing research is limited to exploring the mechanism of interaction between GOI and green innovation performance, and does not form a relatively complete GOI model of “GOI -> innovation behavior -> CEP”. There remain few studies that have confirmed that the green competitive advantage formed by GOI and green innovation can improve CEP. Chinese enterprises are in the transition stage of economic development, and enterprises have begun to change their concepts and began to pay attention to the importance of environmental protection, but relevant theoretical and empirical studies are scarce in China. Therefore, the internal mechanism between GOI and CEP needs to be further explored.

### 2.3. Government Environmental Regulation (GER)

Government environmental regulation (GER) proposed by Eiadat et al. [18] is a series of environmental policies which implemented by the government in order to promote environmentally innovative behavior of the organization and reduce the influence of the organization on the environment. As one of the effective measures of environmental management, a great number of studies demonstrate that government environmental regulation can improve green innovation performance by facilitating organizational inputs in environmental research and development, and effectively weaken the negative influence of enterprise production on the environment by reducing energy consumption and pollutant emission. 

As one of the effective means of environmental management, the GER that effectively reduce the negative impact on the environment by promoting organizational green innovation is under global scholars’ concern [19]. Although the influence of GER on organizational innovation was examined by a large number of studies, the relationship between them remains unclear [4,14]. Some studies showed that GER positively affect innovation performance [19], others show that GER is negatively associated with innovation performance [20]. In addition to these two views, some scholars proposed that the effect of GER on innovation performance is uncertainty [16]. The inconsistency of the relationship between GER and innovation leads to the inconsistency of the relationship between other pressures and CEP. While most scholars strive to verify the relationship between GER and innovation, Ramanathan et al. [4] constructed a theoretical model and discussed the moderation effect of GER on the relationship between sustainability innovation and CEP. However, the existing studies have not explored that whether GER affects the relationship between sustainable innovation and CEP taking GER as an external factor. Hence, the main aim of our study is to explore the heterogeneity of the relationships between different intensity of GER and different categories of innovation behaviors. In order to further reveal the moderation effect of GER on the indirect impact of GOI on CEP via SER and SEI, this study conducts a moderated mediation model and aims to provide new insights into understanding of the action mechanism of GOI on CEP. 

### 2.4. The Impact of GOI on SER and SEI

The organizational identity theory holds that as a method or a real phenomenon that reflects organizational characteristic, organizational identity can strengthen the influence of organizational goal, motivate members to shape organization-oriented behaviors, and urge the organization members to pay more attention to organizational goal [21]. Members who understand and accept the organizational identity and their behaviors are highly consistent with organizational behaviors [22]. GOI incorporates environmental problems into organizational identity and stresses organizational attention to environmental problems. Any behavior related to environmental management deserves vigorous encouragement and support [9,10]. GOI has a positive influence on environmental behavior by prompting members to perceive and ponder on environmental problems. As environmental laws and regulations are continuously improved and consumers’ environmental awareness is constantly enhanced, corporate managers will strive to establish GOI, and make organizational members understand and accept it. In this case, GOI makes organizational members aware that environmental protection can bring corporates green competitive advantages. Encouraged by GOI, members take an active part in corporate behaviors related to environmental management [23]. 

As an innovation model capable of balancing corporate development and environmental protection, SER and SEI not only stress on helping enterprises build a competitive advantage through exploration innovation and exploitation innovation, but also emphasizes pollution prevention, lowering environmental cost of product lifecycle and sustainable development [3,24]. When motivated by GOI, members are bound to take an active part in exploration innovation-related behaviors such as new design and creating new product, technology and knowledge, and they will also actively participate in utilization innovation-related behaviors like upgrading existing technology and product, improving product and service quality or enhancing existing function [13]. Moreover, motivated by GOI, members will have a positive cognition on emission pollution prevention (reduce the emission of sewage, exhaust gas, and waste), product management (reduce environmental cost of product lifecycle), sustainable development (weaken the influence of corporate growth on the environment), and members are willing to take part in these behaviors. In light of the above analysis, GOI mobilizes members to participate in SER and SEI, and increases the efficiency and effectiveness of corporate sustainable exploration and utilization innovation. Therefore, this study proposes the following hypotheses:

**Hypothesis 1** **(H1a):**
*GOI is positively associated with SER.*


**Hypothesis 2** **(H1b):**
*GOI is positively associated with SEI.*


### 2.5. The Impact of SER and SEI on CEP

According to the explanation of Maletič et al. [11] about these two innovation models, this study contends that they can influence CEP from two aspects: innovation and sustainability. From the perspective of innovation, SER takes exploration innovation as the main innovative behavior and creates new management knowledge system and new market by putting available resources into creating new eco-friendly products and technology which are is conducive to reduce the negative impact on the environment [12]. Meanwhile, SEI takes exploitation innovation as the main innovative behavior and emphasis on the improvement of existing knowledge and technologies, and expanding the application scope of existing products and services, which increases the efficiency of existing distribution channels to improve the efficiency of environmental management [25]. Hence, SER and SEI contribute to the improvement of CEP from the perspective of innovation.

From the perspective of corporate sustainability, SER and SEI emphasize pollution prevention, green innovation of products and technology, and the ability to achieve harmonious development among the economy, society, and ecological environment [26]. These two innovation models paying close attention to overall management innovation are devoted to green development and environmental protection [11]. The representative corporate behaviors which are focused and implemented by SER and SEI include weakening the influence of product cycle on ecological environment through process and technology innovation, reducing material consumption and increasing waste recycling rate by improving environmental technology and operation flow, and establishing cultural atmosphere and social responsibility of recognizing sustainable development within the enterprise by developing knowledge and skills of members [25]. Therefore, we propose the following hypotheses:

**Hypothesis 3** **(H2a):***SER significantly and positively affects CEP*.

**Hypothesis 4** **(H2b):***SEI significantly and positively affects CEP*.

### 2.6. The Meditation Effect of SER and SEI

Existing research shows that GOI is able to enhance organizational members’ understanding of environmental and innovation management [13]. Song and Yu also suggest that a higher level of GOI keeps the members highly consistent with the corporate goals and increases the initiative of organization members to proactively solve environmental problems [7]. Hence, with the encouragement of GOI, organizational members positively participate in environmental protection behaviors, such as controlling pollutant emission, increasing the efficiency of energy utilization, improving product quality and weakening the influence of product on environment by promoting the SER and SEI. Given that SER and SEI are likely to increase the effectiveness and efficiency of environmental management CEP via positive innovation management and the improvement of corporate sustainability [3,11], we expect that GOI strengthen corporate green competitive advantage and improve CEP through increasing the efficiency and effectiveness of SER and SEI enhance corporate sustainability. Therefore, we hypothesize: 

**Hypothesis 5** **(H3a):***SER positively mediates the relationship between GOI and CEP*.

**Hypothesis 6** **(H3b):***SEI positively mediates the relationship between GOI and CEP*.

### 2.7. The Moderation Effect of GER on The Mediating Role of SER and SEI

GER contains environmental standards and organizational willingness to confront dynamics, which is the link between CEP and innovation [27]. Previous research in GER suggested that more strict and forward-looking environmental standards were proposed to promote corporate environmental management at a higher level of GER [4,27]. Regarding the strict GER, enterprises are more likely to make a change to avoid excessive environmental cost. Different from SEI, SER leads to innovation results greatly improved, which can help enterprises approach and even go beyond the strict and forward-looking environmental standards set by the government. Meanwhile, strict environmental standards play a crucial role in motivating enterprises to implement SER by reducing the uncertainty and the cost of SER. At higher levels of GER, enterprises are more willing to implement SER, and the effectiveness and efficiency of SER are improved by strict and forward-looking environmental standards. Therefore, when GER is at a higher level, SER is more beneficial to the improvement of CEP. Given the indirect of GOI on CEP via SER, we expect that GER positively moderates the mediation effect of SER on the link between GOI and CEP.

When GER is at a lower level, the lack of strict and forward-looking environmental standards lead to a result that enterprises are unwilling to make a great change [28]. The existing research indicated that SEI effectively reduce the consumption of material, water and energy, and constantly improve the production efficiency by improving existing technologies and products, improving product and service quality or improving existing function [11]. Compared with current product and production process, the outcome of SEI (e.g., product innovation and process innovation) improved to some extent meet the requirement of low-level government regulation for environmental protection [12]. Meanwhile, SEI characterized by fewer risk and lower cost cope with uncertainties brought by the lack of strict environmental management standards [29]. Therefore, when GER is at a lower level, SEI is more beneficial to the improvement of CEP. Given the indirect of GOI on CEP via SEI, we expect that GER negatively moderate the mediation effect of SER on the link between GOI and CEP. Hence, we specify the moderated mediation model and hypothesize that: 

**Hypothesis 7** **(H4a):***The mediating effect of SER on the relationship between GOI on CEP is stronger when GER is high. Specifically, SER will strongly mediate this indirect effect at higher levels of GER than at lower levels*.

**Hypothesis 8** **(H4b):***The mediating effect of SEI on the relationship between GOI on CEP is stronger when GER is low. Specifically, SEI will strongly mediate this indirect effect at lower levels of GER than at higher levels*.

Figure 1 illustrates the relationships between GOI, SER, SEI, GER, and CEP.

## 3. Research Methods

### 3.1. Samples and Data Collection

To test our hypotheses, we applied a questionnaire survey method by collecting survey data from Chinese enterprises. The sample is randomly selected from the central region, the eastern region, the northeastern region and the western region of China. In order to obtain the real and reliable data of this study, we taken the Chief Executive Officer (CEO) and the managers of environmental protection or Research and Development (R&D) departments as the respondents of the questionnaires. To increase efficiency the questionnaire survey, the members of the research team have confirmed that each respondent is willing to accept the questionnaire survey before we mailed the questionnaires.

We adopted the questionnaire items used by the existing studies to ensure the rationality of the questionnaire structure. Meanwhile, we applied the back translation method to translate the original items to Chinese and modified some items according to Chinese context. After we completed the preliminary design of the questionnaire, we have asked six experts to modify some ambiguous or incorrect items. Then, we invited 10 top-level managers from Wuxi, Nanjing, and Suzhou to fill in the questionnaire and identify issues and ambiguities in items. 

A total of 700 questionnaires were sent, and a total of 585 questionnaires were finally collected. The response rate is 83.57%. To obtain valid questionnaires from the returned questionnaires, we eliminate invalid questionnaires according to the following principles. Principle 1: the respondents are not the CEO and the managers of environmental protection or R&D departments. Principle 2: three reverse items are set in the questionnaire, and the absolute value of the reverse items is greater than 3. Principle 3: the results of the questionnaire present appear obvious regularity. Principle 4: the questionnaire is not fully completed. Therefore, 205 invalid questionnaires were removed, among which 16 questionnaires were not completed, 56 questionnaires showed obvious regularity, 82 questionnaires were not filled by the CEO or the managers of environmental protection or R&D departments, and 51 questionnaires ignored three reverse items. Finally, 380 valid questionnaires were finally obtained resulting in an effective rate was 54.28%.

To check the possibility of common method variance (CMV), Harman single factor test was conducted using SPSS 20.0 (IBM, Armonk, NY) in this study [30,31]. The results of exploratory factor analysis (EFA) related to GOI, GER, SER, SEI, and CEP showed that there are six factors and the first factor only explained 14.06% of total variance. According to relevant research [32], we used *t*-tests to compare the general characteristics and model variables between first 25% responses and late 25% response to check the non-response bias. The results showed that there is no significant differences between early responses (25%) and late responses (25%). Therefore, it is unlikely that no-response bias significantly affect the data and results of this study. Table 1 shows the characteristics of the sample.

### 3.2. Measures

Measures of green organizational identity (GOI), government environmental regulation (GER), sustainability exploration innovation (SER), sustainability exploitation innovation (SEI), and corporate environmental performance (CEP) in this study were adapted from previous studies and Likert-type scales anchored at 1 = strongly disagree to 7 = strongly agree. 

(1) Dependent variable: Corporate environmental performance (CEP) was measured by four items from previous empirical studies [12]. CEP [3] is used to evaluate the efficiency of company’s material and energy consumption (e.g., “The efficiency of the consumption of raw materials has improved during the last three years”). 

(2) Independent variables: According to previous studies [9,13], green organizational identity (GOI) refers to “an interpretive scheme about environmental management and protection that members collectively construct in order to provide meaning to their behaviors”. GOI [9] was measured by five items adapted from existing studies (e.g., “high-level, middle-level managers and employees have a strong sense of mission to corporate management and environmental protection”).

(3) Mediating variables: Sustainability exploration innovation (SER) and sustainability exploitation innovation (SEI) are two main forms of sustainability-orientation innovation, which is used to assess the implementation of a company’s sustainability-related innovation [11]. 

Following to existing research [12], SER was measured as a higher-order construct composed of sustainable product and process development (SPPD) and sustainability-oriented learning (SOL). SPPD and SOL were respectively measured using four items developed by [3]. SPPD was used to refer to the green process engineering and product innovation (e.g., “The organization makes improvements to radically reduce environmental impacts of products and services’ life-cycles”). SOL refers to the developing capabilities and competence for sustainability-related innovation (e.g., “The organization is characterized by a learning culture stimulating innovation for sustainability”). Sustainability exploration innovation (SEI) was measured using six items (e.g., “We make use of appropriate tools and techniques to improve the stability of key production processes”) [12].

(4) Moderating variable: Government environmental regulation (GER) refers to a series of government environmental policies to reduce enterprises’ environmental impact and encourage enterprises to engage in environmental innovation [1,18]. In this study, GER was assessed using the four items (e.g., “government environmental laws that impact their companies are effective in tackling environmental problems directly”) [18].

(5) Control variables: A large number of studies have certified that firm size and age have significantly effect on environmental management behavior [33]. To ensure that the moderated meditation model proposed in this study is robust, two firm-level variables were added to the model. According to the measurement method used by previous research [30], the firm size measured by the total number of employees and the firm age measured by the total number of years since a company has been established were controlled.

### 3.3. Instrumentreliability and Validation

Following previous research [12], this study measures SER as a single second-order construct consisting of two sub-constructs termed: Sustainable product and process development (SPPD and sustainability-oriented learning (SOL). As shown in Table 2, SPPD and SOL reflect the second-order construct (SER). 

To evaluate the reliability, we estimated the factor loading of each item. The results showed that the factor loading of SEI6 is lower than 0.5. To increase the reliability, we deleted SER6 and refined the measures from the dataset of this study [8]. As Table 3 shows, the Cronbach’s alphas for all measurement scales range from 0.841 to 0.900. The value of each scale is higher than the recommended cut-off value of 0.70. Therefore, all measurement scales have adequate internal reliability.

To evaluate convergent validity of the scales, we compare the values of factor loadings, composite reliabilities, and average variance extracted with the recommended threshold values suggested by [8,34]. As presented in Table 3, the factor loadings of measurement scales range from 0.724 to 0.844, and the value of factor loadings for each scale is above the recommended threshold value of 0.70. The composite reliabilities (CR) of all measurement constructs ranges from 0.853 to 0.900, and the value of CR for each measurement scale is above the recommended threshold value of 0.80. The average variance extracted (AVE) of all constructs range from 0.549 to 0.651, and the value of AVE for each construct is above the recommended threshold value of 0.50. The convergent validity of all constructs was supported by the values of factor loadings, CR, and AVE, which are above the cut-off values. 

To evaluate the discriminant validity, some previous studies proposed the criteria: the square root of VAE of a latent variable should exceed the correlation coefficient between the rest latent variables [32,34,35]. Table 4 shows the mean, the standard deviation (SD), the square of the AVE, and correlations between the variables. As presented in Table 4, the square root of the AVE of each latent variable is above the correlations between respective paired constructs, in support of discriminant validity. 

To evaluate the construct validity of measurement model, we conducted confirmatory factor analysis (CFA) using MPlus 7.0 (Muthén & Muthén, Los Angeles, CA). The CFA results show a good fit (χ^2^/df = 1.337, Comparative fit index (CFI) = 0.981, Tucker-Lewis index (TLI) = 0.978, Root mean square error approximation (RMSEA) = 0.030) and support the construct validity of latent variables. 

## 4. Results

### 4.1. Results of Direct and Mediating Effects

Following Schnettler et al. [36], structural equation models (SEM) were performed using MPlus 7.0 to test the hypotheses. Firm size and firm age were controlled in the structural model consisted GOI, SER, SEI, and CEP, and the goodness-of-fit indices showed a good fit with the data (χ^2^/df = 1.987, RMSEA = 0.051, CFI = 0.943, and TLI = 0.936). The results for H1a, H1b, H2a and H2b are shown in Figure 1, and the meditation results for H3a and H3b are shown in Table 5.

As shown in Figure 2, the path coefficient between GOI and SER is statistically significant (*β* = 0.771, *p* < 0.01), in support of H1a that SOI is positively associated with SER. Similarly, the path coefficient between GOI and SEI is also statistically significant (*β* = 0.641, *p* < 0.01), which confirms H1b that GOI is positively associated with SEI. Further, the results support H2a that SER has a significant and positive influence on CEP (*β* = 0.292, *p* < 0.01). The results also support H2b that SEI has a significant and positive influence on CEP (*β* = 0.302, *p* < 0.01).

To estimate the indirect effects of GOI on CEP via SER and SEI, we specified 1000 bootstrapping iterations at 99% confidence interval. When the 99% confidence CIs (lower and upper) do not contain zero, the hypothesis that the mediating effect is significant is supported [35,36]. As presented in Table 5, the results of structural model indicate that the meditating role of SER in the relationship between GOI and CEP is significant because the 99% confidence interval does not contain zero (lower = 0.061, upper = 0.389). Hence, H3a is supported. Similarly, the results indicate that the 99% confidence interval does not contain zero (lower = 0.089, upper = 0.298) and confirm that SEI also has a significantly and positively mediating role between GOI and CEP. Therefore, H3b is supported. 

### 4.2. Results of Moderated Mediation Effects

Following previous study [35], we employed the PROCESS proposed by Hayes [3] using SPSS 20.0 to test the moderated mediation effect of GER on the indirect effect of GOI on CEP via SER and SEI. Then, we specified a moderation mediation model that estimates the indirect effect of X (GOI) on Y (CEP) via M (SER or SEI) at different levels of V (GER). The previous literature [37,38] suggested that the moderating effect of M on the indirect of X is significant when the moderating effects of M are different at different levels of the moderator. In this study, we defined low level (−1SD) when GER equaled one standard deviation below the mean, and defined high level (+1SD) when GER equaled one standard deviation above the mean to check the moderating effect of GER. As suggested by Hayes [3], Model 14 of the PROCESS was performed. The results for H4a are shown in Table 6, and the results for H4b are shown in Table 7.

The results presented in Table 6 show that the indirect effects of GOI on CEP via SER (lower = 0.064, upper = 0.224) is statistically significant when GER is high (−1SD). The indirect effects of GOI on CEP via SER (lower = 0.075, upper = 0.330) is also statistically significant when GER is high (+1SD). Hence, the results indicate that the indirect effects of GOI on CEP via SER increases (boot effect increases from 0.136 to 0.179) when GER is from low level (−1SD) to high level (+1SD), in support of H4a that GER positively moderates the indirect effect of GOI on CEP via SER. 

The results presented in Table 7 show that the indirect effects of GOI on CEP via SEI (lower = 0.090, upper = 0.269) is statistically significant when GER is high (−1SD). Then, the indirect effects of GOI on CEP via SER (lower = 0.045, upper = 0.243) is also statistically significant when GER is high (+1SD). Therefore, the results confirm that the indirect effects of GOI on CEP via SEI decreases (boot effect decreases from 0.173 to 0.132) when GER is from low level (−1SD) to high level (+1SD), in support of H4b that GER negatively moderates the indirect effect of GOI on CEP via SEI. 

## 5. Discussion

More and more attention has been paid to the environmental management and sustainable innovation of enterprises in developing countries, especially the environmental behavior and sustainable innovation behavior of enterprises in China which is a major country of energy consumption and pollutant emission [1,30]. Therefore, it is particularly important to optimize government environmental management and enterprises’ behavior related to environmental management to improve environmental performance and realize green development. Previous studies emphasized the promoting effect of GOI and GER on corporate innovation and environment performance [1,7]. However, the relationship between GOI and CEP, as well as the role GER, remains controversial. This study provides new insights regarding the moderation effect of GER and the mediating role of SER and SEI to reveal the association between GOI and CPF in the Chinese context.

Our findings showed that the completely mediating effects of SER and SEI on the relationship between GOI and CEP were supported, nevertheless, the direct effect of GOI on CEP was not found. This study provided more empirical evidence for the results that indicate GOI is associated with green innovation, which was assessed by some previous studies [7,10,13], especially SER and SEI proposed by Maletič et al. [11]. Moreover, our study extends some studies that aims to reveal the consequences of GOI by proposing and confirming that GOI promote CEP via SER and SEI, respectively [9,10,13]. Taking the mediation role of SER and SEI into account, enterprises who want to improve CEP in developing countries should not only emphasize their green innovation behaviors (SER and SEI), but also establish GOI. 

The moderating role GER of in the indirect effect of GOI on CEP via SER and SEI was confirmed by the results of the moderating mediation effect model. Different from some prior scholars who strive to reveal the driving effect of GER on innovation and performance [1,4,14], our study aims to examine the moderating effect of GER on the association between GOI and CEP through SER and SEI from the perspective of moderating effect. The results show that GER significantly moderate the relationship between SER and CEP, as well as the relationship between SEI and CEP. The level of GER is associated with the level of standards and the willingness of enterprises to manage their own environment behaviors, which affects the effectiveness and efficiency of SER and SEI [27]. Therefore, the relevant studies on GER should not only pay attention to the direct or indirect effects of GER as antecedent variable of environmental performance and green innovation behavior, but also pay attention to the moderating effects of GER on other influence paths.

Most previous studies related to GER sought to test the Porter hypothesis regarding the relationship between regulations, innovation, and performance simultaneously [4,39], but a very few studies pay attention to the moderating role of GER. Regarding the moderating effect of GER on the mediation effects of SER and SEI, our findings revealed that the moderating effects of GER on the mediating role SER and SEI in the relationship between GOI and CEP are different. GER positively moderated mediating effect of SER on the relationship between and CEP. On the contrary, GER negatively moderate the mediating effect of SEI on the relationship between GOI and CEP. This suggests that enterprises in developing countries should choose innovative models between SER and SEI to improve the effectiveness and efficiency of environmental management according to the level of GER.

## 6. Conclusions

Focusing on environmental management of enterprises in developing country, this study proposed a moderating meditation model to reveal the moderation effect of GER on the indirect effect of GOI on CEP via SER and SEI. 

Our findings provides new insights into understanding the effect of GOI on CEP via SER and SEI. In China, SEI is more conducive to improving CEP, which is different from previous research conclusions that SER is more conducive to improving CEP [12]. One possible explanation is that China is in the initial stage of economic transformation, and Chinese companies may be more willing to implement the exploitation innovation characterized by low lower cost and lower risk. Therefore, the efficiency and effectiveness of SEI is better, which is more conducive to the improvement of CEP. On the contrary, enterprises are not willing to engage in the exploratory innovation with higher cost and higher risk, which leads to the result that SER is not as good as SEI in improving CEP.

The results of this study further enrich the research on GOI proposed by Chen [9]. Our findings indicate that GOI has different impacts on different innovation behaviors, and further demonstrates that Chinese companies that establish GOI are more willing to conduct SER. In China’s manufacturing industry, GOI is more conducive to the implementation of SER than SEI. Therefore, in the process of promoting CEP through GOI, top managers need to further subdivide the sustainable innovation behavior, and need to pay attention to selecting SER and SEI. The reason for this result may be that a company with high level of GOI is willing to form a strong initiative in environmental protection through seeking SER. This further enriches the research on the relationship between GOI and enterprise innovation behavior. 

Furthermore, the results of moderating mediation model presented in our study revealed that GER positively moderate the mediation effect of SER, and negatively moderate the mediation effect of SEI. The results of the study indicate that in a centralized state such as China, GER is an administrative tool that has a significant impact on corporate environmental management. Our findings provide some guidance for the formulation of environmental regulations in China. CEP can be more effectively improved by adjusting the intensity of GER. Due to the great difference in the innovation ability of enterprises in different regions in China, policy makers can adjust the intensity of GER according to according to the innovation capacity and types of enterprises in different regions. Thereby, it is a better way for promoting the development of China’s economic transformation by improving the environmental performance of Chinese enterprises. In addition, facing the different intensity of GER, enterprises conduct a dynamic approach to improve their environmental performance by allocating the innovation resources between SER and SEI. This provides empirical evidence for exploring the potential influence of GER on positively environmental management, which was suggested by Ramanathan et al. [4]. 

Our studies still suffers from several limitations, which are considerably worth to be further explored in the future research. One of limitation of this study is that merely analyzes the mediating role of SER and SEI respectively. Previous studies related to ambidextrous innovation confirmed that firms need to optimize the balance between exploration and exploitation by the aid of internal and external contexts [40]. Therefore, sustainability ambidextrous innovation studies are needed, given that the interaction or balance between SER and SEI may affect the relationship between GOI and CEP.

The second limitation is that this study focuses on the moderation role of GER without considering its driving role in enterprise innovation and performance. The positive effect of GER on innovation performance has been confirmed [6,41], and the moderating role of GER is proposed by Ramanathan et al. [4] and examined by our studies. This means that GER is likely to influence innovation performance and environmental management via the direct influence and the moderation effect, simultaneously. Hence, the relationship between GER and enterprise environmental management needs to be further revealed regarding the direct effect and moderation effect of GER.

## Figures and Tables

**Figure 1 ijerph-16-00921-f001:**
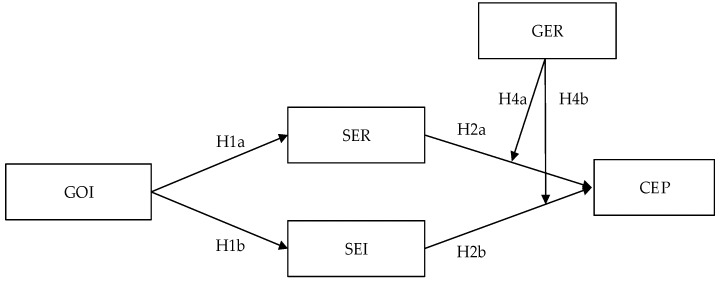
Conceptual framework of the relationships between green organizational identity (GOI), sustainability exploration innovation (SER), sustainability exploitation innovation (SEI), government environmental regulation (GER), and corporate environmental performance (CEP).

**Figure 2 ijerph-16-00921-f002:**
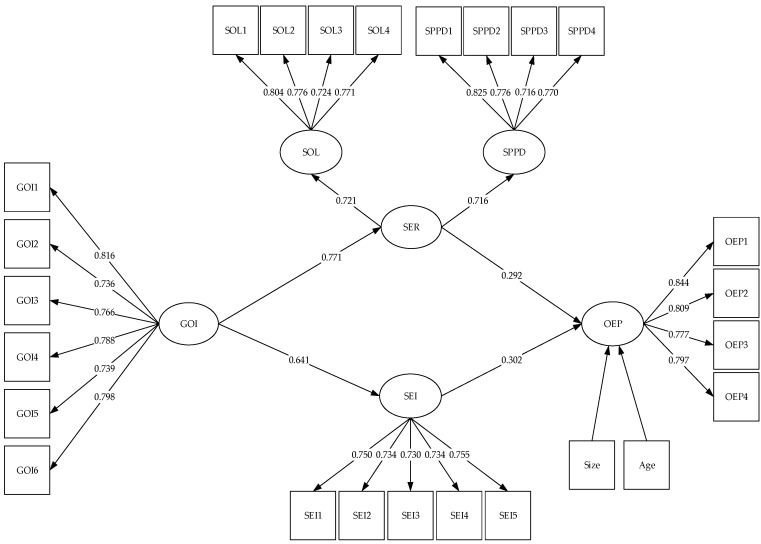
Results obtained from the structural mode including green organizational identity (GOI), sustainability exploration innovation (SER), sustainability exploitation innovation (SEI), and corporate environmental performance (CEP), SPPD: Sustainable product and process development, SOL: Sustainability-oriented learning.

**Table 1 ijerph-16-00921-t001:** Characteristics of the sample (*N* = 380).

Variable	Indicators	Number	%
Status of holding	Collective-holding	112	29.47%
	Private-holding	84	22.11%
	State-holding	109	28.68%
	State-owned	22	5.79%
	Foreign-holding	53	13.95%
Firm size (Number of employees)	<100	70	18.42%
	100–500	134	35.26%
	501–1000	61	16.05%
	>1000	115	30.26%
Firm age (year)	<5	28	7.37%
	6–10	78	20.53%
	11–15	58	15.26%
	16–20	77	20.26%
	>20	139	36.58%

**Table 2 ijerph-16-00921-t002:** Second-order construct of sustainability exploration innovation (SER).

First-Order Construct	First-Order		Second-Order Construct
	Indicator	Loading	Loading
SER with SPPD	SPPD1	0.825	0.716
	SPPD2	0.776	
	SPPD3	0.716	
	SPPD4	0.770	
SER with SOL	SOL1	0.804	0.721
	SOL2	0.776	
	SOL3	0.724	
	SOL4	0.771	

SPPD: Sustainable product and process development. SOL: Sustainability-oriented learning.

**Table 3 ijerph-16-00921-t003:** Reliability values and factor loadings for scales’ items.

Construct	Factor Loadings ^a^	CR & AVE
Corporate environmental performance (CEP) (α = 0.841)		CR = 0.882 AVE = 0.651
CEP1	0.844	
CEP2	0.809	
CEP3	0.777	
CEP4	0.797	
Sustainability exploration innovation-SPPD (SPPD) (α = 0.852)		CR = 0.855 AVE = 0.597
SPPD1	0.825	
SPPD2	0.776	
SPPD3	0.716	
SPPD4	0.770	
Sustainability exploration innovation-SOL (SOL) (α = 0.850)		CR = 0.853 AVE = 0.592
SOL1	0.804	
SOL2	0.776	
SOL3	0.724	
SOL4	0.771	
Sustainability exploitation innovation (SEI) (α = 0.855)		CR = 0.859 AVE = 0.549
SEI1	0.750	
SEI2	0.734	
SEI3	0.730	
SEI4	0.734	
SEI5	0.755	
SEI6 ^b^		
Green organizational identity (GOI) (α = 0.900)		CR = 0.900 AVE = 0.600
GOI1	0.816	
GOI2	0.736	
GOI3	0.766	
GOI4	0.788	
GOI5	0.739	
GOI6	0.798	
Government environmental regulation (GER) (α = 0.854)		CR = 0.854 AVE = 0.595
GER1	0.776	
GER2	0.793	
GER3	0.746	
GER4	0.769	

CR: Composite reliability. AVE: Average variance extracted. ^a^ All item loadings are significant at *p* < 0.01. ^b^ SEI6 is deleted because that the factor loading is lower than 0.5.

**Table 4 ijerph-16-00921-t004:** Descriptive statistics and discriminant validity.

Variable	Mean	SD	1	2	3	4	5	6	7
1. Age	1.187	0.312							
2. Size	2.654	0.732	0.508 **						
3. GOI	5.323	0.999	0.135 *	0.074	(0.775)				
4. SEI	5.325	0.899	0.040	0.072	0.536 **	(0.741)			
5. SER	5.427	0.837	0.006	0.071	0.550 **	0.640 **	(0.771)		
6. GER	5.612	0.932	0.129 *	0.029	0.590 **	0.420 **	0.414 **	(0.771)	
7. CEP	5.357	1.002	0.049	0.529 **	0.529 **	0.579 **	0.579 **	0.333 **	(0.807)

Two-tailed tests significance at * *p* < 0.05, ** *p* < 0.01. Diagonal values in bold represent the square root of the AVE. SD: Standard deviation. GOI: Green organizational identity. SER: Sustainability exploration innovation. SEI: Sustainability exploitation innovation. GER: Government environmental regulation. CEP: Corporate environmental performance.

**Table 5 ijerph-16-00921-t005:** Results for the indirect effects of GOI on CEP via SER and SEI.

Hypothesized Indirect Effects	Specific Indirect Effect of GOI on CEP	Bootstrapping Percentile 99% CI			Conclusion
		Lower 0.50%	Boot Estimate	Upper 0.50%	
H3a	CEP SER GOI	0.061	0.225	0.389	Supported
H3b	CEP SEI GOI	0.089	0.194	0.298	Supported

GOI: Green organizational identity. SER: Sustainability exploration innovation. SEI: Sustainability exploitation innovation. CEP: Corporate environmental performance.

**Table 6 ijerph-16-00921-t006:** Conditional indirect effect of GOI on CEP via SER at specific levels of GER.

Hypothesis	Moderator (V)	Indirect effect of GOI on CEP via SER			Conclusion
		GOI (X) -> SER (M) -> CEP(Y)			
		Boot Effect	Lower 0.5%	Upper 0.5%	
H4a	Low: GER (−1SD)	0.136	0.064	0.224	Supported
	High: GER (+1SD)	0.179	0.075	0.330	

GOI: Green organizational identity. SER: Sustainability exploration innovation. CEP: Corporate environmental performance. GER: Government organizational regulation.

**Table 7 ijerph-16-00921-t007:** Conditional indirect effect of GOI on CEP via SEI at specific levels of GER.

Hypothesis	Moderator	Indirect effect of GOI on CEP via SEI			Conclusion
		GOI (X) -> SEI (M) -> CEP(Y)			
		Boot Effect	Lower 0.5%	Upper 0.5%	
H4b	Low: GER (−1SD)	0.173	0.090	0.269	Supported
	High: GER (+1SD)	0.132	0.045	0.243	

GOI: Green organizational identity. SEI: Sustainability exploitation innovation. CEP: Corporate environmental performance. GER: Government organizational regulation.

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
