# Peer review of "The Relationship between Green Organization Identity and Corporate Environmental Performance: The Mediating Role of Sustainability Exploration and Exploitation Innovation"

_ijerph, 2019, doi:10.3390/ijerph16060921_

Reviewer 1 Report

This paper uses a survey method to investigate the role of GER, GOI, SER and SEI in enterprise environmental management regarding the moderation effect of government regulations. The results provide an understanding of the mechanisms by which enterprises could activate the positive effect of GOI and CEP to balance between economic growth and eco-environmental protection. The paper deals with an important issue in many developing counties and proposes an innovative model to improve the effectiveness and efficiency of environmental management of enterprises.

The major contribution I find is the literature review and empirical analysis. However, a careful English editing is needed to improve the quality of this paper. For example, P1, line 16~17, there is a gramma mistake. P1, line 40, “efficiently” should be “efficiency”. There are many more grammar problems. Some clarification questions remain as follows:

1.     P1, line 37~39: Can you provide an example for GOI, GER, SER and SEI respectively (one example for each)?  It is difficult for readers to distinguish these technical terms without real-life examples.

2.     P2, Line 15: What is “GOU”? Why “GOI” is more likely to improve SER and SEI?

3.     P6, line 10: Why the mediating effect of SEI is stronger “when GER is high low”? Is there is a typo here?

4.     P6, line 15: How is your random sampling is conducted? What is your population? How your samples are drawn from the population?

5.     P6, line 16: Can you explain why there are 4 regions in your sample? Why do you choose them as your sampled regions?

6.     P6, line 31: What is the meaning of “three reverse items”?

7.     P6, line 22: What is the meaning of principle three? There must be a mistake in grammar in the sentence: “The questionnaire present appear obvious regularity”.

8.     P6, line 33: There is a typo in principle 4. The questionnaire is “not” fully completed. The word “not” should be deleted.

9.     P6, line 38~45: Please explain why you need to use CMV?

10.  P8, line 5: Please explain how you can construct SER with SPPD and SOL.

11.  P12, line 6: what is “environmental behavior”? It is a rather vague terminology. Is it relevant here?

Author Response

Response:

We appreciated that reviewers could propose some sincere comments which is benefit for us to improve our study. Following these comments, we made many major to revisions, and invited a native English colleague to check our manuscript. Here, we offer a brief overview of our revised manuscript.

 1. The specific explanations of GOI, GER, SER, and SEI are reflected in the research review. The explanation of GOI can be found on page 3, line 19 to line 20. The explanation of GER can be found on page 3, line 43 to line 46. The explanation of SER can be found on page 2, line 46 to line 50, the explanation of the SEI can be found on page 3, line 1 to line 4.

 2. The GOU should be misspelled and has been modified to GOI.

Following another reviewer’s comments and suggestions, we readjusted the structure of the article, and we deleted “GOI” is more likely to improve SER and SEI”. The changed content can be found on page 2, line 19 to line 23.

In addition, the reason why GOI is more likely to improve SER and SEI can be found in section 2.4. the Impact of GOI on SER and SEI (Hypothesis 1a and hypothesis 2 on pager 4, line 18 to line 47 ).

 3. This should be a misspelling. We have made changes. Correctly stated as“Hypothesis 8 (H4b). The mediating effect of SEI on the relationship between GOI on CEP is stronger when GER is low” which can be found on page 6, line 20.

 4. This study uses the questionnaire survey method. The research data mainly comes from the central, eastern, northeastern and western regions, with a total of 2 municipalities and 12 provinces. The research object is an independent company that does not include molecular institutions or subsidiaries. The survey targets are mainly senior managers of the company and managers of R&D departments. They adopt a combination of on-site investigation and postal research.

 5. This is a common method used by academics and Chinese officials (including the Chinese Bureau of Statistics) to distinguish China’s different regions. The purpose of this study was to enable samples from different regions to prevent samples from coming from the same region, resulting in non-general and universal results.

 6. In the general questionnaire, the Likert 7 scale is used, and the values 1-7 are set, which represent “very disagree”, “disagree”, “general”, “a little agree” and “very agree”. In this way, the attitude and willingness of the respondents are investigated, but all the items are set in the order of 1-7, and the respondents may have the inertia of thinking or fail to accurately reflect the true attitude of the respondent. Therefore, sometimes the order of the options set by the questionnaire is reversed, and the values 1-7 are set to represent “very agree”, “somewhat agree”, “general”, “disagree”, “very disagree”.

 7. In the survey, we usually set according to the Likert five-level scale, setting the values 1-7, which represent “very disagree”, “disagree”, “general”, “a little agree” and “strongly agree”. The respondents choose according to the meaning of the question and their own attitude, but sometimes some of the respondents are not able to fill out the questionnaire according to their true wishes, or they are unwilling to continue filling after filling in for a period of time. In the case of random selection of answers, for example, a large number of consecutive "general" cases will appear. Such questionnaires are called "regularity questionnaires", and these questionnaires are invalid and must be eliminated.

 8. Thanks for your suggestion, it has been removed.

 9. The common method bias refers to the artificial covariation between the predictor and the criterion variable caused by the same data source or scorer, the same measurement environment, the project context, and the characteristics of the project itself. The common method bias is widely used in the study of psychology and behavioral research by using the questionnaire method. Each questionnaire in this study was completed by the same respondent. We told the investigators to have time separation (cannot complete the questionnaire in a short continuous time), but we cannot guarantee that each person is investigated. Those will be on time, so before we conduct data analysis, we need to check whether there is a common method bias and eliminate the systematic error.

 10. According to the concept and measurement of SER, SER is a latent multidimensional constructs composed of two single-dimensional constructs (SPPD and SOL). Building a Latent Factor Multidimensional Construct (SER) consists of two steps. In the first step, the first-order factor SPPD was constructed using the observable indicators SPPD1, SPPD2 SPPD3 SPPD4, and the first-order potential factor SOL was constructed using the observable indicators SOL1, SOL2, SOL3, and SOL4. In the second step, two second-order potential factors, SPPD and SOL, are used to construct the second-order factor SER. This study uses MPlus 7.0 to implement SER. The results are shown in the figure below.                                           

The index results are shown in Table 2.

Table 2. Second-order construct of Sustainability Exploration Innovation (SER).

First-order construct

First-order

Second-order construct

Indicator

loading

loading

SER with SPPD

SPPD1

0.825

0.716

SPPD2

0.776

SPPD3

0.716

SPPD4

0.770

SER with SOL

SOL1

0.804

0.721

SOL2

0.776

SOL3

0.724

SOL4

0.771

 11. Thank you for your question. Environmental behavior refers to the general term for the activities of environmental legal relations including international organizations, countries, enterprises, institutions and individual citizens to directly and indirectly influence the environment. According to the activity mode, it is divided into environmental behaviors and non-action behaviors. The former refers to the activities that the main body should engage according to the needs of environmental protection. The latter refers to the activities that the environmental law requires the subject to consciously suppress. Taboo behavior. The green vocabulary of the key vocabulary mentioned in this paper is also an important part of environmental behavior. This vocabulary is a high-frequency vocabulary in Chinese literature, and foreign experts may not be familiar with this vocabulary.

Reviewer 2 Report

In this paper, literature review and statistical methods are used adequately. This reviewer offers 8 questions to improve the article:

a) In the theoretical section, the authors acknowledge that some of the proposed relationships have been previously analyzed. It would be interesting to explain in more detail why it is interesting to replicate the study in an emerging economy such as China and its particularities from the point of view of the environment. This argument would better justify the validity of the study.

b) Regarding the relationships between variables, the recent literature has analyzed that

different types of regulation (GER)  influence differently in innovation, whether it is end-of-pipe or preventive regulation. And it could have been dealt with in greater depth in the paper.

c) The hypotheses, following the first argument, should be contextualized to the case of an economy such as China. In the way that they are presented, they are generalizations, some of which are previously analyzed.

d) At the end of the hypothesis, it would be recommended to represent a model with the relationships between variables because otherwise it is difficult for the reader to understand it.

e) The high response rate for a random sample is surprising. Could you explain something more about how this rate has been achieved?

f) This reviewer considers that the constructs used have been measured differently in other studies. In particular, this reviewer has doubts about whether a CEP can be measured exclusively by efficiency in the use of materials and energy. Why have not other business performance measures been included?

g) In terms of formal issues, paper should be reviewed. There are several mistakes. On page 2, authors write GOU instead of GOI. In page 10, they write SOI instead of GOI. On page 2, proposal is repeated. And this reviewer has found many more mistake. In-depth review please.

h) Poor conclusions. It is practically a copy of the discussion. Instead of repeating what has been found, the consequences of these findings should be explained in the context of China.

Author Response

Response:

Thank you for reading and commenting on the paper. We have re-adjusted the paper again. The specific modification are as follows.

a) I am very grateful to the reviewers for their comments. In the theoretical section we explain why the importance of replicating a study in China. You can check details in Page1,line 37 to line 42 and Page2,line 1 to line 42.

 b) Thank you for your suggestion, you made a good idea and your suggestion is very pertinent. We also realize that this is a very interesting research question.

The existing literature related to GER focuses on the effect of the type and intensity of GER on entrepreneurial innovation and enterprise performance. However, there are still two unsolved problems. One problem is that the effects of the different types and intensity of GER on entrepreneurial innovation have not been explored. Another problem is that the effect of GER on different innovative behaviors of enterprises has not been solved, and there is also a lack of relevant literature. Hence, the main aim of our study is to explore the heterogeneity of the influence mechanism of GER on different categories of innovation behaviors.

If we consider the type and intensity of GER at the same time, it will lead to the workload of the paper is too large, and it may be difficult to answer this question in detail. Therefore, in the following research, we will focus on the impact of different types of GER on different types of innovation behaviors, and strive to get a deeper understanding of the relationships between different GERs and entrepreneurial innovation behaviors.

 c) The reviewer gave us a very meaningful suggestion. This study lacks links with the actual situation of the Chinese economy. Following to the reviewers' suggestions, we added content connected with the actual situation of the Chinese economy in the section Introduction, Literature Review and Hypotheses, and Conclusion of our research to highlight the value of this study. You can check it on page 2, page 3, page 4 and page 14.

 d) Thanks for your pertinent suggestion. We have represented a conceptual model with the relationships, and you can check it on page 6, line 22 to line 24.

 e) Chinese companies are generally reluctant to participate in unpaid research activities. In order to ensure the efficiency of sample collection, our team selects the number of enterprises in the selected area to send mail or telephone communication before the questionnaire is issued, indicating to the purpose and part of our investigation, after confirming that they are willing to participate in the questionnaire. Our team will issue the questionnaire again. After 700 companies have decided to participate, we will send a questionnaire. In this way, while ensuring the randomness of the sample, the recovery amount and recovery time of the questionnaire can be improved. Researchers like this are widely used in existing research. (e.g., Peng et al., 2018)

Peng B, Tu Y, Wei G. Can Environmental Regulations Promote Corporate Environmental Responsibility? Evidence from the Moderated Mediating Effect Model and an Empirical Study in China[J]. Sustainability, 2018,10(3):641.

 f) The existing literature also has different views on the way CEP is measured. For example, due to the level limitation, we can't ensure which measurement method is better. We can only find a more suitable method for our research. For example, China has not established such a way, we can not get the environmental performance of Chinese companies in this way. This method has been proven to be feasible and credible by previous literature and has certain credibility.

As for why not choose other business performance. At the beginning of the design of the paper framework, we also conducted related discussions. Considering that the main focus of this study is to test whether the green organization identification in Chinese enterprises can effectively improve the existence path of the enterprise environment and the regulatory effect of environmental regulation on the path. The focus is on answering academic debates about the impact of sustainable exploration innovation and sustainable use innovation on corporate environmental performance. Therefore, this study only selected environmental performance.

We are working on a paper that focuses on testing the Porter hypothesis, considering environmental performance while considering business performance. If possible, we can further examine whether GOI can influence business performance through SER and SEI in subsequent research. We will improve the flow in the next study.

 g) We have carefully examined and modified the form. In addition, we revised the content on page 2, and deleted the repeated content. You can check it on page 2.

 h) Thanks to your comments. According to the reviewer's opinion, we revised the conclusion. We removed the content of the conclusion and the discussion part, and explained it in combination with the situation of Chinese enterprises, highlighting the uniqueness of this research. The details can be found on page 14, line 5 to line 35.

Round  2

Reviewer 2 Report

Authors have introduced all my suggestions. Maybe some part of the text, especially the conclusions could still be improved. However,  I agree with all the changes. Therefore, I give my approval for its publication.